# Better Outcomes for Older people with Spinal Trouble (BOOST) Trial: a randomised controlled trial of a combined physical and psychological intervention for older adults with neurogenic claudication, a protocol

Esther Williamson,[1] Lesley Ward,[1] Karan Vadher,[2] Susan J Dutton,[2] Ben Parker,[3] Stavros Petrou,[3] Charles E Hutchinson,[3] Richard Gagen,[3] Nigel K Arden,[4] Karen Barker,[4,5] Graham Boniface,[1] Julie Bruce,[3] Gary Collins,[2] Jeremy Fairbank,[4] Judith Fitch,[6] David P French,[7] Angela Garrett,[1] Varsha Gandhi,[1] Frances Griffiths,[3] Zara Hansen,[1] Christian Mallen,[8] Alana Morris,[1] Sarah E Lamb[1,2]

For numbered affiliations see end of article.

**Correspondence to**
Dr Esther Williamson;
esther.williamson@ndorms.ox.ac.uk

## ABSTRACT

**Introduction** Neurogenic claudication due to spinal stenosis is common in older adults. The effectiveness of conservative interventions is not known. The aim of the study is to estimate the clinical and cost-effectiveness of a physiotherapist-delivered, combined physical and psychological intervention.

**Methods and analysis** This is a pragmatic, multicentred, randomised controlled trial. Participants are randomised to a combined physical and psychological intervention (Better Outcomes for Older people with Spinal Trouble (BOOST) programme) or best practice advice (control). Community-dwelling adults, 65 years and over, with neurogenic claudication are identified from community and secondary care services. Recruitment is supplemented using a primary care-based cohort. Participants are registered prospectively and randomised in a 2:1 ratio (intervention:control) using a web-based service to ensure allocation concealment. The target sample size is a minimum of 402. The BOOST programme consists of an individual assessment and twelve 90 min classes, including education and discussion underpinned by cognitive behavioural techniques, exercises and walking circuit. During and after the classes, participants undertake home exercises and there are two support telephone calls to promote adherence with the exercises. Best practice advice is delivered in one to three individual sessions with a physiotherapist. The primary outcome is the Oswestry Disability Index at 12 months. Secondary outcomes include the 6 Minute Walk Test, Short Physical Performance Battery, Fear Avoidance Beliefs Questionnaire and Gait Self-Efficacy Scale. Outcomes are measured at 6 and 12 months by researchers who are masked to treatment allocation. The primary statistical analysis will be by 'intention to treat'. There is a parallel health economic evaluation and qualitative study.

**Ethics and dissemination** Ethical approval was given on 3 March 2016 (National Research Ethics Committee number:

## Strengths and limitations of this study

- ► The BOOST (Better Outcomes for Older people with Spinal Trouble) Trial is a large, multicentred, randomised controlled trial with a prespecified sample size estimate and includes health economic and qualitative evaluations.
- ► The primary outcome is the Oswestry Disability Index, but we also collect a range of secondary outcomes including objective physical capacity measures and self-reported pain, symptoms and mobility, which are highly relevant to this patient group.
- ► The intervention is individually tailored and uses group supervision to maximise the potential for cost-effectiveness.
- ► Due to the nature of the intervention, participants cannot be blinded to treatment allocation.
- ► At some sites, the same physiotherapist is delivering both trial interventions, but treatments are delivered according to a manualised protocol and quality control visits are conducted to reduce the risk of introducing bias to the trial.

16/LO/0349). This protocol adheres to the Standard Protocol Items: Recommendations for Interventional Trials checklist. The results will be reported at conferences and in peer-reviewed publications using the Consolidated Standards of Reporting Trials guidelines. A plain English summary will be published on the BOOST website.

**Trial registration number** ISRCTN12698674; Pre-results.

## INTRODUCTION

Neurogenic claudication (NC) is a condition that frequently affects older adults.[1] The

burden of symptoms is substantial. NC presents as pain, discomfort or other symptoms radiating from the spine into the buttocks and legs, which are provoked by walking or prolonged standing and relieved by sitting or lumbar flexion.[2] Other signs and symptoms include weakness, altered sensation, fatigue and gait changes.[2] Pain in the lower back is usual but not a necessary diagnostic feature. The symptoms of NC are thought to arise from pressure on the nerves and blood vessels in the spinal canal caused by degenerative changes narrowing the volume of the spinal canal. Narrowing may or may not be evident on radiological imaging.[2 3] When narrowing is evident radiologically, the condition is termed lumbar spinal stenosis (LSS). The relationship between imaging results and symptoms is inconsistent as not all people with radiological narrowing report symptoms of NC.[2 3]

Symptoms due to spinal stenosis are the most common reason for spinal surgery in people over 65 years of age.[4] However, the effectiveness of surgery is unclear, and it exposes older people to considerable risk of complications, including wound infection and cardiorespiratory problems.[4–6] Surgery is also expensive. Current clinical guidelines suggest that physiotherapy is an option for patients with symptoms arising from LSS before proceeding to surgery.[7] However, we do not know whether physiotherapy is effective, nor which physiotherapy techniques should be used.[8] A Cochrane systematic literature review reports that the current evidence for non-operative care for people with NC is very low to low quality.[9] All recent reviews agree that higher quality trials are needed.[9–13] Despite NC being a condition associated with older age, interventions tested to date have not targeted age-associated changes in the musculoskeletal system of participants (such as generalised sarcopenia and frailty) or the psychological impact of pain. In order to generate high-quality evidence regarding non-surgical care for NC, our aim is to conduct a high-quality, multicentred, randomised controlled trial (RCT) of a physiotherapist-delivered, combined physical and psychological intervention.

## OBJECTIVES
The following are the objectives:
▶ To estimate the clinical and cost-effectiveness of a physiotherapist-delivered, combined physical and psychological intervention for older adults with NC compared with best practice advice.
▶ To explore whether indicators of frailty, behavioural factors and radiological (MRI) biomarkers can identify groups of participants who are more likely to respond positively to the intervention using prespecified subgroup analyses.
▶ To conduct a parallel, longitudinal qualitative study with a sample of trial participants to better understand participant experiences of living and ageing with NC, and to inform implementation if the intervention is successful.

## METHODS/DESIGN
### Overview
The study design is a multicentred RCT with embedded qualitative study and economic evaluation (see figure 1).

We are currently recruiting community-dwelling older adults with symptoms of NC. Recruitment opened on 25 July 2016 and we anticipate recruitment to be completed around June 2018. Participants are identified from National Health Service (NHS) physiotherapy and consultant spinal clinics in community and secondary care settings. In addition, participants are identified through a primary care-based cohort study (the Oxford Pain, Activity and Lifestyle (OPAL) cohort study). The OPAL cohort study is being conducted in the same localities as the trial.

The experimental intervention is a physiotherapist-delivered, combined physical and psychological programme. Participants attend an individual session, followed by 12 group sessions delivered over a 12-week period. During the individual session, participants undergo an assessment and are prescribed the exercises they will carry out during the group sessions tailored to their ability, symptom presentation and general health. The group sessions consist of (1) education and group discussion based on cognitive behavioural (CB) techniques; (2) warm-up and circuit exercises; and (3) a walking circuit. The education component focuses on pain management strategies, engagement with home exercises and increasing physical activity. The exercises target muscle strength, balance and flexibility, while the walking circuit aims to increase walking self-efficacy and mobility. The education component and supervised exercise are provided in groups of approximately six participants to maximise the potential for cost-effectiveness. There are two follow-up phone calls on completion of the group sessions to encourage adherence with the home exercise programme.

The comparator is advice given by a physiotherapist (best practice advice), ideally in one session, but up to two further review sessions are permissible. Advice includes self-management strategies, home exercises and encouragement to increase physical activity.

Participants are randomised in a 2:1 ratio (intervention:control) and followed up for 12 months (primary endpoint).

### Eligibility
Participants are included in the trial if they fulfil the eligibility criteria listed in box 1. In the UK, the majority of adults are registered with a primary care practice. Due to the pragmatic nature of this trial, we include people with symptoms consistent with the clinical presentation of NC rather than a diagnosis of spinal stenosis based on evidence of narrowing of the spinal canal on an MRI scan. NC presents as a cluster of symptoms easily recognised using simple self-report questions identified in a recent systematic literature review[3] (table 1). These questions have excellent sensitivity and specificity for identifying NC[3] and are used to screen for eligible participants.

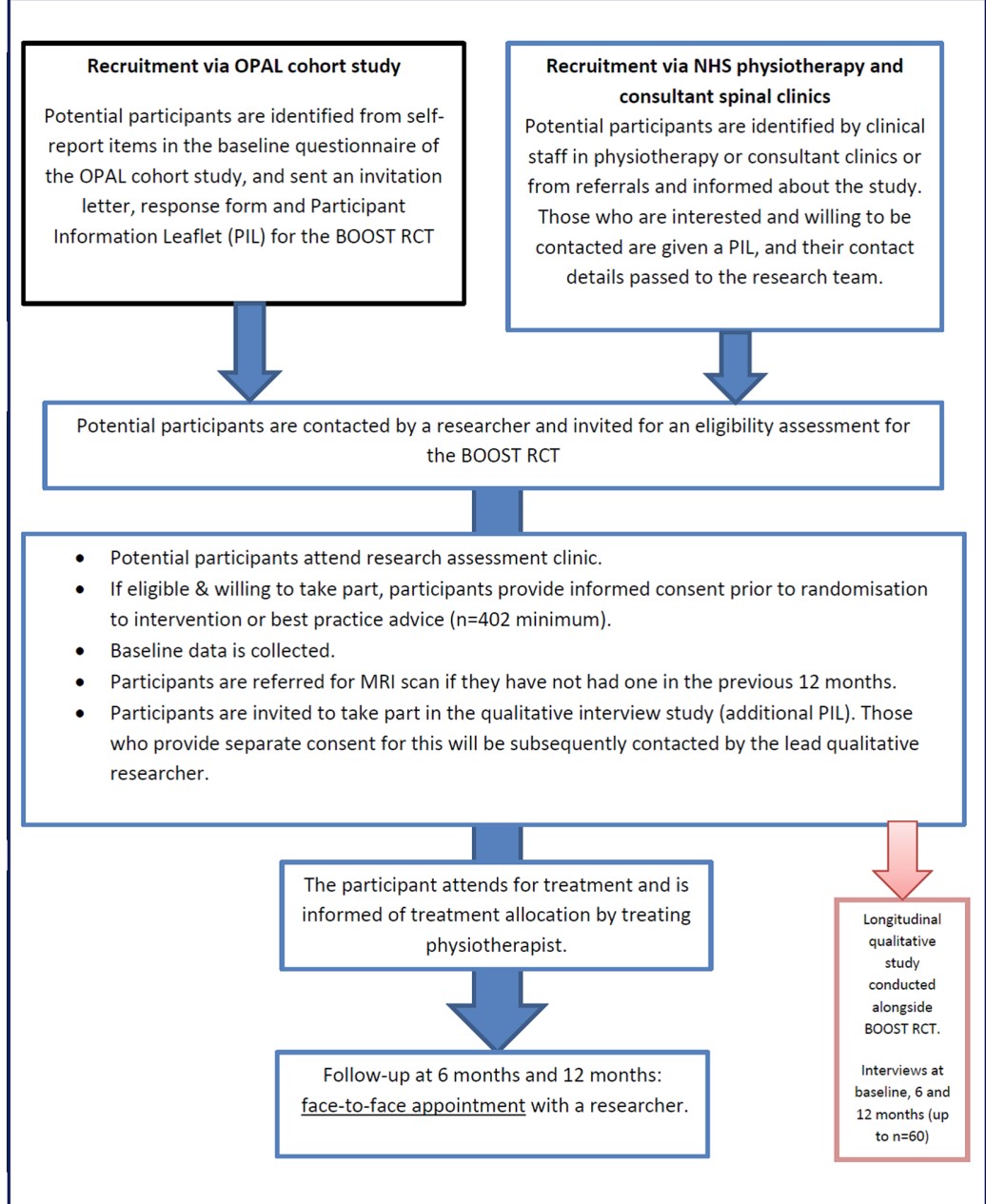

**Figure 1** Study flow chart. BOOST, Better Outcomes for Older people with Spinal Trouble; NHS, National Health Service; OPAL, Oxford Pain, Activity and Lifestyle; RCT, randomised controlled trial.

The exclusion criteria are largely related to the participant being unable to participate in the intervention, for example, if they are unable to follow instructions or mobilise short distances without assistance. Participants are not excluded on the basis of any existing comorbidities unless their general practitioner (GP) feels inclusion in the study places them at risk.

### Approach

Potential participants are approached via two routes:

#### Physiotherapy and consultant spinal clinics in community and secondary care NHS settings

Potential participants are identified by clinical staff in physiotherapy and consultant spinal clinics or from referrals. Staff are asked to identify potentially eligible patients based on age (≥65 years) and symptoms (back and/or leg symptoms) and to screen out those on surgical waiting lists if that information is available. Clinical staff approach potentially eligible patients attending clinics, provide information about the study and ask if they are interested in being contacted by the research team. Clinical staff may also contact new referrals by telephone to inform them about the study. Potential participants who are interested and willing to be contacted by the research team are provided with a participant information leaflet (PIL), and their contact details are passed to the BOOST (Better Outcomes for Older people with Spinal Trouble) researcher for full eligibility screening.

## Box 1 Eligibility criteria

**Inclusion criteria.**
► Registered with a primary care practice.
► 65 years and over.
► Participant is willing and able to give informed consent for participation in the randomised controlled trial.
► Reports symptoms consistent with neurogenic claudication.

**Exclusion criteria.**
► Living in a residential care or nursing home.
► Has a terminal condition with a life expectancy of less than 6 months.
► Any substantial health or social concern that, in the opinion of the patient's general practitioner, would place the patient at increased risk or inability to participate, including known inability to provide informed consent, for example, dementia.
► Unable to walk 3 m (width of a small room) without the help of another person.
► On a surgical waiting list.
► Presents with cauda equina syndrome or signs of serious pathology requiring immediate referral for investigations.
► Cognitive impairment (defined as an Abbreviated Mental Test score of 6 or less).
► Registered blind.
► Unable to follow verbal instructions, which would make participation in the experimental treatment arm of the trial impractical, for reasons including severe hearing impairment not corrected by a hearing aid or inability to follow simple safety instructions (eg, English comprehension).

### The OPAL cohort study

The OPAL cohort study is a population-based cohort study. Participants are identified from a random sample of patients aged over 65 years registered with each participating primary care practice. The OPAL cohort study is currently ongoing at 34 primary care practices, and will be described elsewhere.

Embedded within the cohort study postal questionnaires are self-report questions to identify individuals with possible NC (box 1). During the process of consent for the cohort study, OPAL participants are asked for additional consent for the University of Oxford to provide information and an invitation to clinical trials relevant to their clinical profile. OPAL participants who fulfil the initial criteria for the BOOST Trial (table 1) are invited to take part in eligibility screening for the trial and provided with the BOOST PIL. OPAL participants who accept the invitation for screening are then contacted by telephone for initial screening.

### Eligibility screening

Potential participants identified via NHS spinal clinics or the OPAL cohort study are telephoned by the BOOST researcher (physiotherapists or research nurses) working at each site. During this telephone call, the researcher provides further information about the trial and completes initial eligibility checks. Eligible patients are invited to attend a research clinic appointment for a full assessment. If an individual requests to undertake the initial eligibility check in person, then this is arranged.

The research clinic appointment includes assessment of symptoms to ensure they are consistent with NC (using the questions in table 1), and screening for cauda equina syndrome or signs of serious pathology requiring immediate referral for investigations. Potential participants also undertake the Abbreviated Mental Test (AMT)[14] to screen for cognitive impairment, which would make it difficult for a person to participate in the intervention. The AMT contains 10 items to assess orientation, registration, recall and concentration. This test can be used by any clinician and only takes 3–4 minutes to administer.[15] A score of 6 or below (out of 10) suggests cognitive impairment requiring further assessment and patients are advised to consult their GP.[15–17] A mobility assessment is undertaken if required to ensure the participant is able to mobilise independently at least 3 m unassisted (with or without a walking aid).

### Informed consent

On completion of the full eligibility assessment, eligible participants are asked to provide written informed consent prior to enrolling them into the trial. The consent is taken by a researcher who has completed training in the consent procedures for the BOOST Trial.

**Table 1** Screening questions from the OPAL cohort study questionnaire[3]

| Questions | Response required to be eligible for BOOST Trial |
|---|---|
| 1. In the past 6 WEEKS, have you had back pain *and/or* pain or other symptoms such as tingling, numbness or heaviness that travelled from your back into your buttocks or legs? (Note: If the answer to this question is no, then the participant will not complete the remaining questions). | Yes. |
| 1. Does standing make the pain or symptoms in your buttocks or legs worse? 2. Does walking make the pain or symptoms in your buttocks or legs worse? 3. Does sitting down make the pain or symptoms in your buttocks or legs better? 4. Does bending forward (eg, to push a shopping trolley) make the pain or symptoms in your buttocks or legs better? | Yes to at least one of these questions. |

BOOST, Better Outcomes for Older people with Spinal Trouble; OPAL, Oxford Pain, Activity and Lifestyle.

## Baseline assessment

After providing consent, the participant then completes a baseline questionnaire and undergoes a clinical assessment by the researcher. Data collection is described in table 2. There are a number of variables only collected at baseline for the purposes of providing descriptive data on the sample. The participant is weighed using digital scales wearing light, indoor clothing with their shoes removed. Weight is recorded to the nearest 0.1 kg. Height is measured using a stadiometer positioned against a wall. The participant stands on the platform, shoes removed, as upright as possible, hands by their side. The head plate of the stadiometer is lowered until it gently rests on the top of the participant's head and the height is recorded in metres from the measuring rod, to the nearest 0.001 m (1 mm).

The participant provides self-reported data including the demographic variables listed in table 2, comorbidities including other pain problems (measured using the Nordic Pain Questionnaire[18 19]) and their current mobility status. Measures of mobility status include use of walking aids inside and outside, and self-rated walking speed.[20] Change in mobility in the last year is measured using a 5-point scale constructed for the trial.

The STarT Back Screening Questionnaire[21] is completed, allowing participants to be categorised according to their risk (low, medium or high) of developing persistent, disabling symptoms.[21] Self-reported psychological factors with a potential impact on outcome are also collected. These include their confidence to exercise (Exercise Self-Efficacy Scale (short version)[22]) and their intention to carry out their home exercises using a question constructed for the trial (see table 2). Participants' attitudes to the physical changes associated with ageing are measured using the Attitudes to Aging Questionnaire—physical changes subscale.[23]

Variables collected at follow-up are described in the Outcome measures section.

## Imaging

Alongside the RCT, there is an exploration of whether MRI scan parameters along with other baseline factors moderate response to physiotherapy treatment. Indirect visualisation by MRI is the gold standard for diagnosing LSS when a patient presents with NC, and is always undertaken before surgery, but not necessarily before conservative treatment. Increasingly, GPs have open access to MRI, and if MRI were predictive of response to conservative treatment this could aid GPs' clinical decision making. Despite the expense, there is remarkably little evidence about whether MRI scans can guide treatment choice effectively. Research evidence indicates that the fit between symptoms and MRI changes is poor.[24 25] As MRI is currently the most common imaging investigation used, MRI data will be collected for all participants and we will systematically quantify the imaging characteristics. Pre-existing scans, taken in the 12 months preceding

randomisation, will be used where possible to reduce the need for scanning.

Participants will be referred for an MRI study of the lumbar spine if they have not had one in the 12 months prior to randomisation. For these participants, the MRI scan will be taken after completion of other baseline data collection, and where possible before randomisation. Due to the nature of spinal stenosis, we would not expect spinal parameters to change markedly over a year-long period, hence the rationale for including existing scans. Practically, it is not possible to collect all MRI data prebaseline data collection as this may delay treatment and create unacceptable waiting times. For the subset of people who have MRI scans prior to randomisation, we will undertake formal subgroup analysis. We will explore other aspects of the relationship between functional outcomes and scan characteristics in additional analyses (not to be reported alongside the main trial results).

Consent for referral for a new MRI or use of an existing scan is obtained at the time of consent for the trial. Existing scans are transferred to a central repository for analysis in DICOM (Digital Imaging and Communications in Medicine) format.

The MRI data collection follows the protocol typical of NHS imaging departments. This is very similar across departments and efforts have been made to standardise the protocol where significant differences were identified.

We anticipate that a small number of participants will not have an MRI scan due to contraindications or by personal choice. Lack of an MRI scan does not exclude participants from the trial.

## Imaging protocol

The MRI scan is performed supine, with the knees supported in flexion by a small foam wedge, resulting in relaxation of the normal lumbar lordosis. Imaging is performed using a dedicated spine phased array coil.

T1-weighted and T2-weighted sagittal imaging is followed by T2-weighted axial imaging of at least the lower three discs. The axial imaging is either taken as three separate blocks, each angulated and entered on the discs, or as a single block extending from L3 to S1.

Imaging parameters should be near those described in table 3. The BOOST Trial radiologist liaises with site to ensure data scans are suitable for data collection.

MRI scans are assessed by a single observer blinded to treatment allocation. Measurement of bony canal and dural sac cross-sectional area at each vertebral level allows assessment of central canal stenosis. The size of the lateral recess and neural exit foramen is measured and recorded quantitatively. The exact degree of narrowing to confirm stenosis is not well defined. In a review by Steurer et al,[26] a dural sac cross-sectional area of less than 100 mm$^2$ was considered diagnostic of central canal stenosis. Similarly, lateral recess depth and foraminal diameter measurements of less than 3 mm have been considered diagnostic of lateral recess and foraminal stenosis, respectively.

**Table 2** Data collection and outcomes for the BOOST Trial

| Method | Domains measured | Measure | Time points (months) |
|---|---|---|---|
| Participant-completed questionnaire | Demographic information | Age and sex. Current alcohol and smoking behaviour.[78] Ethnicity. Relationship status. Postcode. Type of housing. Current occupation. Education. Unpaid/paid carer (Question: Do you have an unpaid carer? (someone who is not paid to care for you and without whose support you cannot cope; this could be a partner, family member or friend) Yes/No. Paid carer question is the same format.). Household income. | 0 |
| | Back pain and leg symptoms | Oswestry Disability Index (V.2.1a)[28]—primary outcome. | 0, 6, 12 |
| | | Troublesomeness of back and leg problems.[79] | 0, 6, 12 |
| | | Perceived ability to self-manage their condition (Question: We would like you to think about how you are managing your symptoms and your ability to walk and be mobile. How well do you feel that you are managing your back and leg problems TODAY? (VAS: 0=not managing at all; 10=managing extremely well)). | 0 |
| | Quality of life | Euroquol 5 Dimension 5 Level Scale (EQ-5D-5L).[30] | 0, 6, 12 |
| | Other pain | Nordic Pain Questionnaire.[18 19] | 0 |
| | Comorbidity | Self-report of current health conditions. | 0 |
| | Frailty | Tilburg Frailty Index.[37] | 0, 6, 12 |
| | Physical activity | Two items from the Rapid Assessment Disuse Index[80] ((1) time spent moving around on your feet; (2) time spent sitting). | 0, 6, 12 |
| | Mobility | Change in mobility in the last year (Question: Compared with 1 year ago, how would you rate your walking in general? Much better now than 1 year ago; somewhat better than 1 year ago; about the same; somewhat worse than 1 year ago; much worse now than 1 year ago). | 0 |
| | | Self-rated walking speed.[20] | |
| | | Use of walking aids inside and outside (Question: Do you use a walking aid (eg, walking stick, walker) to walk around outside/inside? Yes; no; sometimes). | |
| | | Change in mobility in the last 6 months (Question: Compared with 6 months ago, how would you rate your walking in general? Much better now than 6 months ago; somewhat better than 6 months ago; about the same; somewhat worse than 6 months ago; much worse now than 6 months ago). | 6, 12 |
| | Balance and falls | Prevention of Falls Network Europe self-report of falls and fall-related injuries.[38] | 0, 6, 12 |
| | Self-efficacy | Single item from the Modified Gait Self-Efficacy Scale (10-item)[32] (Question: How much confidence do you have that you would be able to safely walk a long distance such as 1/2 mile? (VAS: 0=no confidence; 10=complete confidence)). | 0, 6, 12 |
| | | Exercise Self-Efficacy Scale (short version).[22] | 0 |
| | | Self-efficacy recovery and maintenance related to performing home exercises.[33 34] | 6, 12 |
| | Exercise adherence | Intention to carry out home exercises[81] (Question: As part of the BOOST Trial, the physiotherapist will ask you to exercise at home at least twice a week for up to 20 min. How much do you agree with this statement? I intend to do these exercises at least twice a week for up to 20 min. Strongly disagree; disagree; somewhat disagree; neither agree nor disagree; somewhat agree; agree; strongly agree). | 0 |
| | | Self-report of adherence to home exercise programme (Question: In the past 6 months, on average, how many times per week have you managed to do your exercises for at least 20 min? Never; 1 day per week; 2 days per week; 3–4 days per week; 5–6 days per week; every day). | 6, 12 |
| | Habit (automaticity) | Index of Habit (short version).[35] | 6, 12 |
| | Fear avoidance | Fear Avoidance Beliefs Questionnaire.[31] | 0, 6, 12 |
| | Beliefs about ageing | Attitudes to Aging Questionnaire—physical changes subscale.[23] | 0 |
| | Global Rating of Change | Change in back and leg problems.[29] | 6, 12 |
| | Satisfaction | Satisfaction with the exercises, changes in back and leg problems, increases in physical activity (All questions follow this format: How satisfied are you with the exercises that you were given to help with your back and leg problems? (VAS 0–4; 0=very dissatisfied; 4=very satisfied)). | 6, 12 |

Continued

| Method | Domains measured | Measure | Time points (months) |
|---|---|---|---|
| Clinical interview and assessment | Height | Measured using a stadiometer. | 0 |
| | Weight | Measured using digital scales. | 0 |
| | Spinal parameters | Sagittal alignment of the spine measured using C7 to wall measure.[40 41] | 0, 6, 12 |
| | Frailty | Hand grip strength[44] measured using a Jamar Plus+ Dynamometer. | 0, 6, 12 |
| | Mobility | 6 Minute Walk Test.[43] | 0, 6, 12 |
| | Mobility/balance | Short Physical Performance Battery.[82] | 0, 6, 12 |
| | Back pain and leg symptoms | STarT Back Screening Questionnaire.[21] | 0 |
| | | Swiss Spinal Stenosis Scale (symptom subscale).[27 83] | 0, 6, 12 |
| | Medication use | Self-report of medication use. | 0, 6, 12 |
| | Health resource use | Client Service Receipt Inventory.[39] | 6, 12 |
| Imaging | Spinal parameters | MRI scan—use existing scan taken in the last 12 months or referred for scan after randomisation. | Variable |

**Table 2** Continued

BOOST, Better Outcomes for Older people with Spinal Trouble; VAS, Visual Analogue Scale.

## Provision of MRI results to participants

MRI scans requested for the purpose of the trial and not as part of a participant's clinical management are research investigations only. These are collected and assessed by the trial radiologist. If a serious spinal pathology is identified, the participant's GP and/or spinal consultant are immediately informed. If no serious pathology is identified, then scan results will be made available to participants at the end of the study if requested. MRI reports will be sent to each participant's GP or spinal consultant so that scan results are explained to the participant appropriately.

## Randomisation and masking

Following baseline data collection, the researcher uses a web-based service to randomise the participants. During this process, the researcher is not informed of the treatment allocation. Instead, an automated email is sent directly to the physiotherapists who provide the interventions.

The web-based randomisation service is provided by the Oxford Clinical Trials Research Unit consistent with UK Clinical Research Collaboration (UKCRC) approved standard operating procedures, ensuring prospective registration and allocation concealment. Randomisation is stratified by centre, age (65–74 years and 75+ years) and gender. Participants are randomised in a 2:1 ratio (intervention:control) to ensure that there are enough participants to run a group intervention and minimise waiting times.

Physiotherapists delivering the interventions and participants cannot be masked to treatment allocation. All participants receive an initial 1-hour appointment. For those randomised to the BOOST programme, this is an assessment prior to attending the group sessions. For those randomised to the control arm, it is their initial physiotherapy session to deliver best practice advice. During this appointment, participants are informed of their treatment allocation by the physiotherapist. To ensure that researchers collecting follow-up data remain masked to treatment allocation, physiotherapists and participants are asked not to share information about treatment allocation with researchers.

The trial statistician and the research staff undertaking quality assurance checks and the qualitative study are not blinded to treatment allocation. The remaining members of the trial management team, including all those who are involved in data management, are masked to treatment allocation.

## Outcome measures

Follow-up data are collected at 6 and 12 months after randomisation, at a clinic appointment. The outcomes are listed in table 2.

## Primary outcome

The primary outcome is low back pain disability measured using the Oswestry Disability Index (ODI V.2.1a)[27 28] at 12 months after randomisation. The ODI is quite widely

**Table 3** Imaging parameters

| Sequ | FOV | Slice | Gap | TR | TE | ETL | Phase | Freq | Nex |
|---|---|---|---|---|---|---|---|---|---|
| T2 sagittal | 370 | 13/4 | 1 | 4061 | 102 | 23 | 320 | 512 | 3 |
| T1 sagittal | 370 | 13/4 | 1 | 446 | 11 | 23 | 224 | 416 | 3 |
| T2 axial | 200 | 30/4 | 1 | 4955 | 111 | 25 | 224 | 320 | 3 |

ETL, echo train length; FOV, Field of view; Freq, Frequency; Nex, Number of excitations; Sequ, Sequence; TE, time to echo; TR, Time to repeat.

used as a measure for NC and very widely used in the field of back pain. A comparison of the psychometric properties of four of the most promising self-report measures for NC demonstrated that the ODI had superior properties to other measures.[27] It is highly applicable to NC because it includes items on standing and walking. Scores range from 0 to 100, with higher scores indicating greater disability. Participants are asked to consider back and leg symptoms when responding, including discomfort, heaviness, aching, tingling and numbness. Responses are not limited to the impact of back pain only.

### Secondary outcomes

A range of self-reported and physical measures are collected to evaluate the impact of the intervention on key treatment targets (symptoms of NC, mobility, physical activity, strength, balance, frailty and falls, and cognitive and behavioural factors related to adherence with exercise and improving physical activity levels).

### Self-reported measures

Self-reported measures related to symptoms and their impact include the Swiss Spinal Stenosis Questionnaire,[27] Global Rating of Change,[29] satisfaction with treatment using a 5-point scale constructed for the trial, health-related quality of life measured using the Euroquol 5 Dimension 5 Level Scale (EQ-5D-5L),[30] and how well participants are managing their leg and back symptoms on a 10-point scale constructed for the trial.

Information is collected about cognitive and behavioural factors targeted during the intervention that are hypothesised to mediate the effects of the intervention. Fear avoidance is measured using the Fear Avoidance Beliefs Questionnaire.[31] Self-efficacy is measured from different perspectives. Participants rate their confidence to walk half a mile using a single item from the Modified Gait Self-Efficacy Scale.[32] To understand maintenance of exercise and physical activity, drawing on theoretical and empirical literature on this topic,[33] participants also rate their confidence to restart their exercises having stopped them (self-efficacy recovery[33 34]) and their confidence to maintain their exercises in the long term (self-efficacy maintenance[33 34]). Measures related to the adherence of exercises and increasing physical activity are collected using the Index of Habit,[35] self-reported frequency of exercise and satisfaction with their attempts to increase their physical activity[36] measured on a 5-point scale constructed for the trial. Change in mobility in the last 6 months is measured using a 5-point scale constructed for the trial.

A range of measures are collected to capture constructs related to ageing. Frailty is measured using the Tilburg Frailty Index,[37] and information about falls and fall-related injuries is collected as recommended by the Prevention of Falls Network Europe.[38] Health resource use will be collected using the Client Service Receipt Inventory.[39]

### Physical assessment

A measure of postural alignment is undertaken to quantify the degree of thoracic kyphosis.[40 41] The participant removes their shoes and socks and stands as upright as possible, with their sacrum and back against the wall, with hands by their sides. The researcher measures the distance from the spinous process of the seventh cervical vertebrae to the wall using a ruler. It is an alternative to the occiput to wall measurement, but reflects kyphosis better as it minimises error due to head position.[40 41]

We then collect measures related to mobility, balance and strength, which are important targets of the intervention and markers related to ageing and frailty.

The Short Physical Performance Battery (SPPB)[42] measures three aspects of physical performance: standing balance, walking speed and the time taken to perform five chair stands. An overall score is given by adding the scores for each test. Researchers follow published guidance on the test which is, briefly, as follows[42]:

### Standing balance

Standing balance is rated on a scale of 0–4 according to the participant's ability to maintain three test positions (side-by-side stance, semitandem and full tandem) for 10 s.

### Walking speed

Walking speed is measured on an eight-feet long walking course with no obstructions for a further two feet at each end. The participant is instructed to 'Walk to the other end of the course at your usual speed, just as if you were walking down the street to go to the shop'. The time taken for the participant to walk between the two markers is recorded to the nearest 0.1 s. The test is carried out twice and the faster of the two times is used to score the test on a scale of 0–4.

### Chair stands

The participant sits in a straight-backed chair with their arms folded across their chest. They are given the following instructions: 'Now stand up straight 5 times in succession, as fast as you can'. The time taken to perform the five chair stands (from the initial sitting position to the final standing position at the end of the fifth stand) is used to score the test on a scale of 0–4. If the participant is unable to complete the test, then they are given a score of 0.

The 6 Minute Walk Test (6MWT)[43] measures the distance that the participant is able to walk in 6 minutes. The researcher marks out an indoor walking course which is flat and straight and marked with cones at each end. The length of the test track is standardised at each site to ensure that the follow-up assessments are carried out on the same length test track. The recommended length of the course is a minimum of 10 m in total, but it is dependent on the space available at each site.

One lap consists of walking to the turnaround point of the course and returning to the start point. All researchers were provided with a 6MWT compact disc which is played during the test and counts down the 6 minutes of the walking test while the researcher counts the number of laps with a lap counter.

Prior to starting the test, the researcher also asks the participant if they have symptoms of NC. If they do not have any symptoms when starting, the participant is asked to verbally indicate if they begin to experience symptoms during the test. The distance at which their symptoms begin is recorded by the researcher.

The researcher measures the participant's hand grip strength[44] using a Jamar Plus+ Dynamometer and follows the protocol outlined by Roberts et al.[45] The participant is seated in a chair with arms, with their hips, knees and ankles at 90°, and their feet flat on the ground. The participant's arm is supported on the armrest with their wrists level with the end of the armrest. During the test the researcher supports the weight of the dynamometer. Using standardised instructions, the participant is instructed to squeeze the handle of the dynamometer until they reach a maximal contraction and hold for 5 seconds. The procedure is repeated on the other side. Three measurements are taken on each hand allowing at least 30 seconds rest between measurements on the same hand. The highest reading is used as the summary measure.

### Follow-up procedures

All participants are invited to attend a face-to face clinic appointment at 6 and 12 months. This is arranged by the researcher at each site. However, if a participant is unable to attend the clinic appointment, they are mailed a questionnaire that contains the primary outcome, all self-reported items and a participant completed version of the Client Service Receipt Inventory, but excludes the physical assessment. If the questionnaire is not returned within 2 weeks, then a second copy of the questionnaire is sent by the BOOST Trial Office as a reminder. If this is not returned within a further 2 weeks, then the BOOST Trial Office carries out a reminder phone call. After another 2 weeks, if the questionnaire has not been returned, then the BOOST Trial Office will attempt to contact the participant by telephone and collect core outcomes consisting of the primary outcome (ODI), pain troublesomeness rating, whether they are on a waiting list for spinal surgery, EQ-5D-5L, self-rated walking ability, falls and falls-related fractures, self-reported exercise adherence, and a brief version of the Client Service Receipt Inventory.

### Adverse events

A safety reporting protocol has been developed to manage the reporting of related and unexpected serious adverse events (SAEs) and directly attributable adverse events (AEs). An AE is any untoward medical occurrence in a participant during a trial. There may or may not be a causal relationship with the trial intervention. AEs may be identified by the physiotherapists delivering the trial treatments or by researchers conducting follow-up assessments who have been trained in reporting procedures. SAEs must be reported to the trial management team within 24 hours of the physiotherapist or researcher becoming aware of the event. The Chief Investigator determines whether AEs require reporting to the Ethics Committee, Data Monitoring and Ethics Committee (DMEC) and Trial Sponsor.

### Training and quality assurance of the research protocol

Researchers undergo approximately 4 hours of training covering eligibility screening, consent taking and data collection. They are provided with a manual containing detailed instructions for all trial procedures. All researchers undergo a quality assurance check to ensure they are following the trial protocol. This involves a BOOST team member observing the researcher carrying out the eligibility screening, taking consent and collecting trial data. Trial paperwork is checked for completeness. If any deviations from the protocol are identified, then further training is provided.

### Study interventions

#### Control intervention: best practice advice

The control intervention is best practice advice which is delivered in a one-to-one session with a physiotherapist. Participants attend an initial appointment of up to 1 hour consisting of an assessment followed by the provision of advice and education. Advice and education includes education about NC, being physically active, use of medications, when to seek more advice and prescription of a home exercise programme (up to four exercises). Flexion and trunk stabilisations are recommended, but the physiotherapist may prescribe other exercises based on their assessment, if required. The physiotherapist may prescribe a walking aid if the assessment indicates (eg, to improve walking by increasing stability or for pain relief). Participants are provided with written information.

Ideally, the control intervention should be delivered in one session. A maximum of two half-hour review appointments is permitted. During these sessions they can re-enforce verbal advice given, and review walking aids or exercises provided in the initial session, but are not permitted to provide treatments such as manual therapy, electrotherapy, acupuncture, hydrotherapy or structured exercise sessions.

The content of the control intervention has been informed by a survey of current physiotherapy practice[46] and through consultation with clinicians and patient representatives. Physiotherapy provision in the NHS is variable for this patient group. Many patients are not referred for physiotherapy, some receive advice on self-management at physiotherapy spinal clinics and some receive a course of physiotherapy comprising advice and exercises. Comer *et al*[8] compared a single advice and education session with up to six sessions of standard physiotherapy and showed no difference in outcomes. We recommend that the majority of participants receive one session of advice and education as no additional benefit has been demonstrated from extra sessions of standard physiotherapy. However, there are situations where the treating physiotherapist will feel that a review appointment is necessary (eg, if they have provided a walking aid and need to review its use) so this is permissible and we felt broadly reflected usual care in the NHS.

### Experimental intervention: the BOOST programme

The BOOST programme will be described in full according to the Template for Intervention Description and Replication guidance[47] elsewhere, including the rationale and development. A brief summary is provided here.

Participants are invited to attend twelve 90 minute group sessions over a 12-week period. We recommend that one physiotherapist delivers the BOOST programme to a group of six participants. If larger groups are conducted, then a physiotherapy assistant or another physiotherapist may be required. Prior to attending the programme, each participant attends an individual appointment (up to 1 hour) for an assessment and to set their individualised exercise and walking circuit targets for the group sessions. The baseline target for the strengthening exercise is tailored to each participant by varying the number of repetitions and sets, and the addition of weights as applicable.

Each session follows the same format. Participants take part in an education and discussion session, facilitated by the physiotherapist (30 minutes) and incorporating behavioural change strategies to encourage adherence with home exercises. This is followed by the exercise programme lasting approximately 1 hour. There is a short warm-up of seated exercises performed as a group, which includes arm raises, trunk rotation, pelvic tilting and knee lifts. Then participants undertake a circuit of strengthening (sitting knee extension, sit to stand, standing hip abduction and standing hip extension), stretching (a combined hip flexor and calf stretch) and a balance exercise.[48 49] Each participant undertakes their individually tailored programme, which is progressed over the 12 weeks. The strengthening exercises are progressed by increasing the number of repetitions and sets, increasing the load or adding speed. The final part of the exercise element is a supervised walking circuit, designed to improve walking ability and fitness,[48] which is also progressed over the 12 weeks by increasing the distance walked, increasing walking speed, adding balance challenges such as stairs or obstacles, or adding weights. The exercises carried out during the supervised sessions make up the home exercise programme (warm-up, exercise circuit and walking).

Participants attend the supervised sessions twice a week for sessions 1–6. As they progress through the programme, attendance becomes less frequent (weekly for sessions 7–9, then fortnightly for sessions 10–12). The home exercise programme is introduced during session 5 so that participants begin to undertake their home exercise programme while supported by the physiotherapist. On completion of the 12 group sessions, participants are asked to carry out their home exercise programme at least twice per week so that it becomes a habitual activity.

The physiotherapist monitors progress during the programme by asking participants to rate how well they feel they are managing their condition (0–10 Numerical Rating Scale) and how their symptoms are affecting walking (walking item from the ODI) at the pregroup assessment, and at sessions 3, 6, 9 and 12. At the end of the 12-week programme, the physiotherapist carries out two follow-up telephone reviews with each participant to promote long-term adherence with the home exercise programme. These take place approximately 1 and 2 months after completing the supervised sessions, and take approximately 15 minutes each.

### Concomitant care

Participants may seek other forms of treatment during the trial if they feel it is necessary. Additional treatments accessed by participants, including contact with their GP or other health professionals, will be recorded on the Client Service Receipt Inventory[39] at follow-up.

### Physiotherapist training and quality assurance of intervention delivery

The interventions are delivered by physiotherapists registered with the Health and Care Professions Council. All physiotherapists delivering the BOOST programme attend a 1-day training course, are provided with an intervention manual and undertake 3 hours of online training. All physiotherapists delivering the control intervention attend 3 hours of training and are provided with an intervention manual. At some sites, the same physiotherapist delivers both arms of the trial. Both interventions are delivered according to a manualised protocol to reduce the risk of introducing bias to the study, and routine quality assurance checks are conducted. Visits are made

to each site and at least one session of each intervention is observed. Feedback is provided to the physiotherapist on completion of the session and any issues or training needs identified. Another visit is arranged if substantial concerns are identified.

A structured checklist is used to monitor intervention delivery and ensure that all elements of the interventions are delivered as intended. We developed the education and discussion session of the BOOST programme with the assistance of a CB therapist who also assisted with training. The CB therapist helped to develop the checklist for the BOOST programme to ensure all the necessary components of the education and discussion section of the session were covered. The checklist is completed during the observed session. The education and discussion session may also be assessed via recording (depending on resources and BOOST staff capacity). All participants provide consent for sessions to be recorded for quality assurance purposes when they enrol in the study, and we seek verbal consent from the physiotherapist.

Following initial quality assurance checks and feedback, we also undertake fidelity assessments of both interventions that are not fed back to the physiotherapists. Feedback at this stage is not provided as we need to understand how this intervention would be delivered in the clinical setting if it were to be implemented.

A structured record of the interventions (treatment log) is completed by the physiotherapists and used to monitor fidelity. We collect attendance rates to monitor adherence with the interventions. Additional site visits will be conducted if any problems with intervention delivery are identified.

## Sample size

At 80% power and 5% two-sided significance levels, the proposed sample size is 321 participants in total providing data at 12-month follow-up (214 in the intervention arm and 107 in the control arm), after which inflation for potential loss to follow-up (20%) yields an overall target of 402 (268 intervention, 134 control). If power is increased to 90%, then a sample size of 429 (286 in the intervention arm and 143 in the control arm) is required, after which inflation for potential loss to follow-up (20%) yields an overall target of 540 (360 intervention, 180 control).

These calculations have been based on the assumption that a between-group difference of five points is considered clinically significant on the ODI, with a baseline SD of 15, consistent with published estimates in older populations and those with NC.[50 51] This yields a standardised difference of 0.33, a moderate effect size, which is consistent with being a reasonable target for a pragmatic trial.[52]

The loss to follow-up of 20% has been based on recent experiences of rehabilitation trials in older adults.[53] We estimate that the therapist effects will be negligible from data that we have generated/published from a series of trials using similar standardised interventions. Our recent trials of hand exercises in rheumatoid arthritis and CB interventions in low back pain generated an intracluster

correlation (ICC) of less than 0.0001.[54 55] We anticipate about 20 therapists delivering the intervention, treating an average of 12–15 participants each. We have not incorporated a formal inflation for a therapist effect as the loss to follow-up allowance is generous and should mitigate against any moderate to large therapist effects.

The sample size is a minimum of 402 participants and a maximum of 540, to be finalised following a review of the sample size assumptions (in particular any evidence of clustering or a larger baseline SD) by the DMEC. A number of assumptions in the sample size estimate will be checked at this interim time point and adaptations made if needed, including the baseline SD of the ODI and the observed ICC. The DMEC will review these assumptions at this time point and make recommendations regarding the final sample size to the Trial Management Group and Programme Steering Committee (PSC). No interim analysis of the primary outcome will be performed.

## Analysis

Data will be reported in accordance with the Consolidated Standards of Reporting Trials guidelines for RCTs and the appropriate extensions.[56] A final statistical analysis plan will be developed by the end of the recruitment period, and we provide an outline description here. The primary analysis will be 'intention to treat', where participants will be included in their randomised groups. Effect estimates together with their 95% CIs will be reported. The primary outcome, ODI at 12 months, will be analysed using a linear multivariable regression multilevel method to take account of any therapist effect and adjusted for the region, baseline ODI score, stratification and important prognostic variables.

Missing data will be minimised by careful data management and training. The nature and mechanism for missing variables and outcomes will be investigated, and if appropriate multiple imputation will be used. Sensitivity analyses will be undertaken, assessing the underlying missing data assumptions.

A secondary complier average causal effect (CACE) analysis[57] will explore the effect of adherence with the intervention (attendance and the participants' engagement with the programme rated by the physiotherapist).[58] For the purposes of the primary CACE analysis, we will define adherence as attending at least 9 out of the 12 sessions (75%). This would ensure that the majority of educational/discussion content is delivered. No one session is considered more important than another regarding educational/discussion content. Core CB concepts are introduced during the earlier sessions and are reiterated during subsequent sessions so attendance at nine sessions would ensure all core content is covered. Attendance at nine sessions will ensure that the participant is introduced to the home exercise programme and has undertaken the exercise programme for a minimum of 6 weeks (sessions 1–9 are delivered over a 6-week period). Six weeks of strength training has been shown to be sufficient to result

in short-term improvements in muscle strength and physical function.[59–61]

We have also defined a priori subgroup criteria based on the published literature and will explore treatment effects by age (65–74 years/75+ years), gender (male/female), Tilburg Frailty Index scores (0–4/5+),[37] Fear Avoidance Beliefs Questionnaire scores (0–14/15+),[62] STarT Back Risk Stratification score (low-risk/medium-risk/high-risk groups),[21] hand grip strength (men: <30/30+; women <20/20+)[63] and SPPB scores (SPPB 0–6 low performance; SPPB 7–9 intermediate performance; SPPB 10–12 high performance[63]). Among participants who have an MRI scan prior to randomisation, we will estimate treatment effects in two subgroups defined by MRI parameters (cross-sectional spinal canal area cut-point of $100\,mm^2$).[26] Subgroup effects will be analysed using interaction with treatment tests and will be displayed using forest plots.[64]

We will carry out a series of additional exploratory subgroup and interaction analyses to identify other MRI scan parameters and baseline factors that predict change in ODI scores between baseline and 12 months. Interaction and polynomial terms will be considered when carrying out the exploratory analysis, and the analyses may be based on continuous or binary cut-points. These models will report variables that predict the outcome at 12 months with 95% CIs and p value. These additional analyses will be presented in secondary publications and with appropriate caveats about the interpretation of exploratory subgroup analyses.

Further supplementary analysis may include mediation analysis to evaluate treatment mechanisms, and exploratory analyses of exercise dose effects including profiling of treatment response trajectories. These are a priori analyses based on the logic model used to develop the intervention, and will examine (1) whether the intervention affects the hypothesised mediators as intended, (2) whether changes in the hypothesised mediators relate to changes in outcomes, and (3) whether the effects of the intervention on the outcomes are attributable to changes in the hypothesised causal pathway.

### Economic evaluation

A prospective economic evaluation, conducted from an NHS and personal social services perspective, is integrated into the trial design. The economic evaluation will estimate the difference in the cost of resources used by participants in the two arms of the trial, enabling costs and consequences to be compared between alternative forms of physiotherapy. The economic assessment method will adhere to the recommendations of the National Institute for Health and Care Excellence Reference Case.[65]

We will estimate the costs of delivering the intervention, including development and training, the cost of delivering sessions, and participant follow-up/management. Broader resource utilisation is captured through two principal sources: (1) participant interview administered at 6 and 12 months postrandomisation and (2) routine health service data collection systems (Hospital Episode

Statistics). Unit costs for health and social care resources will be derived from local and national sources.[66] Costs will be standardised to current prices where possible. Health-related quality of life will be measured at baseline and at 6 and 12 months postrandomisation using the generic EuroQol EQ-5D-5L; national tariff sets will be used to generate quality-adjusted life-years (QALYs).[67–69] We will in the first instance use self-report of the EuroQol EQ-5D-5L measure. Multiple imputation methods will be used to impute missing data and avoid biases associated with complete case analysis.[70] The results of the economic evaluation will be expressed in terms of incremental cost per QALY gained. Non-parametric bootstrap estimation will be used to derive 95% CIs for the mean cost and QALY differences between the trial groups, as well as to populate a cost-effectiveness plane. A series of sensitivity analyses will be undertaken to explore the implications of uncertainty on the incremental cost-effectiveness ratios and to consider the broader issue of the generalisability of the study results. The full details of the economic evaluation will be described in the health economic analysis plan.

### Qualitative study

The aim of the qualitative study is to better understand participant experiences both of living and ageing with NC, and their experience of the interventions delivered during the trial. Understanding the experiences of the participants will inform strategies for implementation if the intervention is clinically effective.

All participants recruited to the trial are eligible. As part of the consent process for the trial, participants are informed about the interview study and asked if they are willing to be contacted by a researcher to receive more information. Participants who agree are provided with an additional information sheet and contacted by the qualitative research team. Prior to starting the first interview, written consent is sought. Consent is reaffirmed verbally prior to each follow-up telephone interview.

We are interviewing participants at three time points over the course of the trial in order to capture physical, psychological, social and contextual changes. Topics explored include current impact of NC on day-to-day life and well-being, beliefs about the role of exercise in ameliorating symptoms, the role of exercise on slowing/reversing physical decline, and how these beliefs impact on adherence to the treatments.

We estimate 60 participants will be required to ensure data saturation is reached in all three interviews while ensuring diversity of participants by age, gender, ethnicity and intervention arm, allowing for attrition over the course of the study.[71 72] In any one recruitment site interview participants are recruited consecutively, and as recruitment proceeds sampling is adjusted to ensure diversity of age, gender, ethnicity and intervention arm.

Interviews are semistructured using prespecified open-ended questions. The interviewer uses prompts to further investigate responses and allows the participant to explore

topics they feel are relevant.[71 73] The first interview takes place at a location convenient to the participant, usually their home. The second and third interviews are telephone interviews to reduce resource implications for the project. However, if a telephone conversation is unfeasible (eg, poor hearing), then subsequent interviews are conducted face to face.

The first interview takes place between randomisation and starting treatment. Participants do not yet know their treatment allocation. Questions focus on the impact of NC on the participant's physical and psychosocial health, their beliefs around exercise and ageing, and concerns and hopes regarding the intervention. The interview takes up to 90 min and is audio-recorded.

The second interview is approximately 1 month after completing treatment. Topics explored include the participant's experiences of the intervention, adherence to home exercises, and any changes in their symptoms, exercise and ageing beliefs, physical activity levels, or life circumstances. The third interview coincides with the 12-month follow-up assessment and further explores these topics, and how they may have changed after an extended period of self-management. Interview schedules are adapted to account for data captured in earlier interviews, and the interviewer has access to the outcome measures for each interviewee for exploration during the interview. The telephone interviews are recorded, and notes are transcribed by the interviewer from the audio recordings.

Audio recordings of first interviews are transcribed verbatim by an independent transcriber, anonymised and allocated an ID number. The telephone interview notes are checked against the audio recordings and linked to the first interview through the ID number. Participants are sent a copy of the transcripts if requested, and may delete any information they would not like to be used.

All transcripts are imported into NVivo and analysed using thematic analysis.[74] Coding is undertaken as each transcript is received. We will undertake cross-case analysis.[75] To understand trajectories of change in relation to back pain and NC, we will undertake longitudinal case comparative analysis, an approach to analysis used previously by the research team.[76]

### Trial management

This trial is run by a UKCRC fully registered clinical trials unit, according to approved and audited standard operating procedures. All trial staff undergo regular training to ensure they are compliant with Good Clinical Practice and other relevant legislation and the requirements such as the Data Protection Act.

### Data management and checking

All data are processed according to the Data Protection Act 1998 and all documents are stored safely in confidential conditions. Each participant is provided with a unique trial identification number. Data are entered manually onto the trial database (OpenClinica). The BOOST Trial Office reviews all data collection forms for completeness and accuracy using automated validation checks, querying missing and nonsensical data with sites, according to trial-specific procedures which have been developed to ensure data quality.

### Patient and public involvement

During the application process for this trial, we assembled a patient and public involvement (PPI) group and we have continued to work closely with them. JuF is the lead PPI representative and a coapplicant and contributed to the design of the trial. We appointed a PPI representative to be an independent member of the PSC. PPI engagement has been undertaken in face-to-face meetings and via emails and phone calls to make it as convenient as possible for the PPI group to contribute. PPI representatives have assisted with the development of the physiotherapy intervention. Two PPI representatives attended the intervention development day along with clinicians and researchers. One PPI representative carried out the proposed exercise programme in her home so she could give feedback on the practicalities of performing the proposed programme. PPI representatives helped us to develop the patient materials for the intervention.

PILs, consent forms and posters advertising the trial have been reviewed by the PPI group and they have provided feedback on the layout and wording. We have piloted questionnaires with our PPI group. PPI representatives have helped with developing interview schedules for the qualitative study, and we have carried out some practice interviews with the PPI representatives prior to undertaking the actual study.

## ETHICS AND DISSEMINATION

Site-specific approvals were provided by the NHS Research and Development Departments at each participating site. The Chief Investigator will submit and, where necessary, obtain approval from the above parties for all substantial amendments to the original approved documents.

There were several ethical issues when designing this study. A study of older adults may identify individuals with previously unidentified cognitive impairment. As part of the screening process, participants complete the AMT. Individuals with a score of 6 or below (out of 10) are excluded as this suggests impaired cognitive function requiring further assessment.[15–17] The researchers, conducting the eligibility screening, are trained to deal with this and to recommend that the person visit their GP to for further assessment.

The study screening procedures may identify individuals who have signs of serious spinal pathology (eg, cauda equina syndrome). In this case, the researcher would discuss it with the participant and as soon as possible with the local principal investigator and/or patient's spinal consultant or GP and take appropriate action.

Some participants will undergo an MRI scan as a research investigation. Some participants may want the

results of their scan. However, as participants would not have had access to a scan as part of their routine NHS care, we will only make results available at the end of the trial unless we detect a serious spinal pathology (eg, spinal malignancy). In this situation, the participant's GP or spinal consultant will be informed immediately. If we were to provide the scan results, this may influence the beliefs of participants and their clinicians about the value of different treatments. This process is made clear to potential participants during the recruitment and consent procedures so participants understand the purpose of the MRI scan.

The PSC provides overall supervision of this research on behalf of the funder. It comprised the Chief Investigator, Project Lead, Trial Manager, Statistician and four independent members (including the committee chair). The PSC monitors trial progress and conduct and provides expert advice. In addition, a DMEC has been appointed. The DMEC consists of three independent experts with relevant clinical research and statistical experience. The DMEC has adopted the DAta MOnitoring Committees: Lessons, Ethics, Statistics (DAMOCLES) charter,[77] which defines its terms of reference and operation in relation to oversight of the trial. No interim outcome analysis is planned. Direct access to research data will be granted to authorised representatives of the Sponsor (University of Oxford), regulatory authorities or the host institution for monitoring and/or auditing of the study to ensure compliance with regulations.

The results will be published in a peer-reviewed journal and at conferences, as well as in a report to the funder. A plain English summary will be made available on the BOOST website for participants (https://boost.octru.ox.ac.uk/).

**Author affiliations**
[1]Centre for Rehabilitation Research, Nuffield Department of Rhuematology, Orthopaedics and Musculskeletal Sciences, University of Oxford, Oxford, UK
[2]Oxford Clinical Trials Research Unit, Centre for Statistics in Medicine, Nuffield Department of Orthopaedics, Rheumatology and Musculoskeletal Sciences, University of Oxford, Oxford, UK
[3]Warwick Medical School, University of Warwick, Coventry, UK
[4]Nuffield Department of Orthopaedics, Rheumatology and Musculoskeletal Sciences, University of Oxford, Oxford, UK
[5]Oxford University Hospitals NHS Trust, Oxford, UK
[6]Patient and Public Involvement Representative, Yorkshire, UK
[7]Manchester Centre for Health Psychology, School of Health Sciences, University of Manchester, Manchester, UK
[8]Arthritis Research UK Primary Care Centre, Institute for Primary Care and Health Sciences, Keele University, Keele, UK

**Acknowledgements** Thank you to the patient and public involvement representatives who have provided advice and feedback on the development of the intervention, patient materials and conduct of the trial.

**Contributors** EW is the lead author of this manuscript. LW, SJD, KV, RG, SP and SEL were involved in writing the manuscript. All authors have read and approved the final manuscript. SEL is the Chief Investigator and the guarantor. EW, FG, SP, CEH, NKA, KB, JB, GC, JeF, JuF, DPF, ZH and CM are coapplicants on the grant awarded by the NIHR Programme Grants for Applied Research (reference: PTC-RP-PG-0213-20002) and were involved in the design of the study and its implementation, as were AG as the trial manager, AM as a research associate, SJD and KV as the trial statisticians, GB and VG as the research physiotherapists, LW as

the postdoctoral researcher responsible for the qualitative study, RG as the clinical research fellow (radiology) and BP as the trial health economist.

**Funding** This research is funded by the NIHR Programme Grants for Applied Research (reference: PTC-RP-PG-0213-20002). Preparatory work for the programme of research was supported by the Collaboration for Leadership in Applied Health Research and Care Oxford at Oxford Health NHS Foundation Trust. SEL and EW receive funding from the Collaboration for Leadership in Applied Health Research and Care Oxford at Oxford Health NHS Foundation Trust and are supported by the NIHR Biomedical Research Centre, Oxford. CM is funded by the NIHR Collaboration for Leadership in Applied Health Research and Care West Midlands, the NIHR School for Primary Care Research and an NIHR Research Professorship in General Practice (NIHR-RP-2014-04-026).

**Disclaimer** The views expressed are those of the author(s) and not necessarily those of the NHS, the NIHR or the Department of Health. The trial sponsor is the University of Oxford. The sponsor has no role in the trial design; collection, management, analysis or interpretation of data; writing of reports; and submission for publication.

**Competing interests** None declared.

**Patient consent** Not required.

**Ethics approval** Ethics approval for the BOOST Trial was given by the National Research Ethics Committee (REC number 16/LO/0349) on 3 March 2016.

**Provenance and peer review** Not commissioned; externally peer reviewed.

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
