## [Reviewer comments · BMJ Open]

ARTICLE DETAILS

TITLE (PROVISIONAL)	Better Outcomes for Older people with Spinal Trouble [BOOST] Trial: A randomised controlled trial of a combined physical and psychological intervention for older adults with neurogenic claudication (protocol)
AUTHORS	Williamson, Esther; Ward, Lesley; Vadher, Karan; Dutton, Susan; Parker, Ben; Petrou, Stavros; Hutchinson, Charles; Gagen, Richard; Arden, Nigel; Barker, Karen; Boniface, Graham; Bruce, Julie; Collins, Gary; fairbank, jeremy; Fitch, Judith; French, David; Garrett, Angela; Ghandi, Varsha; Griffiths, Frances; Hansen, Zara; Mallen, Christian; Morris, Alana; Lamb, Sarah

VERSION 1 – REVIEW

REVIEWER	Carlo Ammendolia University of Toronto, Canada
REVIEW RETURNED	17-Feb-2018

GENERAL COMMENTS	1. a condition specific primary outcome measure would have been more desirable. The condition of interest is NC and this relates to lower extremity symptoms and walking impairment. The ODI does not directly measure these constructs. ODI commonly used in LBP studies but often patients with NC have no back pain. ODI has superior test-retest reliability but not validity. 2. the comparator intervention (1-3 sessions) selected limits conclusions to be made on the effectiveness and cost effective of the Boost program. This comparator is an artificial intervention not commonly seen in real practice. Not likely that a patient would receive one physiotherapy visit/treatment for NC in community setting. A usual care or no treatment arm would have been more informative since it would provide information on the potential benefit of the Boost program beyond usual care or no treatment (that is commonly seen in the real world setting). 3. The 6 minutes walk test has not been validated in this population. An objective valid walking outcome is very important in this population where walking is the dominant functional limitation. 4. The specific details allowing for replication not present in methods section of this protocol. How many participants per group session and how many physiotherapists per group? 5. It is unclear whether the same physiotherapists perform both interventions. If so this is a potential source of bias. 6. Table 2 screening questions may select patients with degenerative LBP and not NC. Eg YES to LBP (no leg symptoms) and LBP better with shopping cart or sitting. By definition NC is specific to buttock and lower extremity symptoms and not LBP. 7. The above issues can be addressed in a sections on potential limitation of the protocol
--

REVIEWER	Birgitta Öberg Department of Medical and Health Sciences Medical faculty Linköping University Linköping , Sweden
REVIEW RETURNED	22-May-2018

GENERAL COMMENTS	This is a protocol that targets an interesting topic where there is an existing knowledge gap. The design is good and the chosen outcomes relevant. One challenge is that the inclusion strategy, as I understand , will lead to inclusion of those who has not primarily contacted the health care because of a heavy burden of their back pain complaints. This means that the population will include both those with minor and major problems. Patients with different feelings of burden will probably have different chances of improvement of the intervention. This can probably be handled in future analysis taking the disability level into the calculations and elaboration of mediators including disability level. It will be of great importance to publish the presentation of the treatment in detail as stated in the protocol
---

VERSION 1 – AUTHOR RESPONSE

Reviewer: 1

Comment: A condition specific primary outcome measure would have been more desirable. The condition of interest is NC and this relates to lower extremity symptoms and walking impairment. The ODI does not directly measure these constructs. ODI commonly used in LBP studies but often patients with NC have no back pain. ODI has superior test-retest reliability but not validity.

Response: Pratt et al [1] compared the ODI with the only condition specific outcome measure available at the time (Swiss Spinal Stenosis Questionnaire) and found no advantage of using it over the ODI. The ODI contains questions pertaining to pain, walking and standing which are very relevant to older adults with NC.

The participants are given the following instructions when completing the ODI to ensure they consider both back and leg symptoms: "pain refers to any symptoms related to your back and leg problems including; discomfort, heaviness, aching, tingling, numbness". This has been clarified in the text.

Comment: The comparator intervention (1-3 sessions) selected limits conclusions to be made on the effectiveness and cost effective of the Boost program. This comparator is an artificial intervention not commonly seen in real practice. Not likely that a patient would receive one physiotherapy visit/treatment for NC in community setting. A usual care or no treatment arm would have been more informative since it would provide information on the potential benefit of the Boost program beyond usual care or no treatment (that is commonly seen in the real world setting).

Response: The content of the control intervention has been informed by a survey of current UK physiotherapy practice for the treatment of NC, as delivered within the NHS [2], and through consultation with clinicians and patient representatives from NHS Trusts throughout England. Physiotherapy provision is variable throughout the UK for this patient group. Many patients are not referred for physiotherapy, some receive advice on self-management at physiotherapy spinal clinics, and some receive a course of physiotherapy comprising of advice and exercises. Comer et al [3] compared a single advice and education session with up to 6 sessions of standard physiotherapy and showed no difference in outcomes. We recommend that the majority of participants receive one

session of advice and education as no additional benefit has been demonstrated from extra sessions of standard physiotherapy. However, there are situations where the treating physiotherapist will feel that a review appointment is necessary (e.g. if they have provided a walking aid and need to review its use) so this is permissible. Our patient representatives and clinical collaborators were in agreement with this decision and felt that it broadly represented usual care in the UK NHS. We have added justification for the control arm.

Comment: The 6 minutes walk test has not been validated in this population. An objective valid walking outcome is very important in this population where walking is the dominant functional limitation.

Response: The reviewer is correct that the 6 minute walk test is not validated in a neurogenic claudication population. However, it has been shown to be a valid measure of mobility and fitness in a variety of conditions including patients with chronic respiratory disease[4], diabetes[5], Charcot Marie Tooth [6], Duchene Muscular Dystrophy [7] and Multiple Sclerosis [8]. The 6 minute walk test was included in a review of measures for patients with chronic pain, fibromyalgia and chronic fatigue disorders, which concluded there was moderate evidence of the reliability, validity and acceptability of this type of testing [9].

From a practical point of view, the test is easy to administer, 6 minutes is not overly demanding of participants who are also being asked to complete other physical tests, and it does not require specialist equipment or a large amount of space for walking track. The group intervention focuses on improving walking distance and speed as well as general fitness and the 6 minute walk test is a suitable way to measure to these domains.

Comment: The specific details allowing for replication not present in methods section of this protocol. How many participants per group session and how many physiotherapists per group?

Response: As noted above, in response to the editor's comment, we have only provided a brief description of the intervention as we have a written separate paper providing a detailed rationale and description of the intervention. This has been submitted to "Physiotherapy" and is currently under peer review. In that paper, a full description of the intervention has been provided, in line with TIDieR guidelines. To include a full description of the intervention in this protocol paper would have made it prohibitively long. We have added some additional details to the text to improve transparency of the intervention.

Comment: It is unclear whether the same physiotherapists perform both interventions. If so this is a potential source of bias.

Response: Yes, at some sites with limited staff availability, the same physiotherapists deliver both interventions. To minimise potential bias, physiotherapists receive separate training for the delivery of each intervention; both interventions are delivered to a standardised, manualised protocol, and we monitor the delivery of both interventions at each site. These details have been added to the text.

Comment: Table 2 screening questions may select patients with degenerative LBP and not NC. Eg YES to LBP (no leg symptoms) and LBP better with shopping cart or sitting. By definition NC is specific to buttock and lower extremity symptoms and not LBP.

Response: We are in agreement with the reviewer. Individuals must report back pain and/or pain or other symptoms such as tingling, numbness or heaviness that travelled from your back into your buttocks or legs to be eligible for the trial. This is stated in Table 2. Those with back pain only are not eligible.

Comment: The above issues can be addressed in a section on potential limitation of the protocol.

Response: Text has been added in the appropriate sections.

Reviewer: 2

Comments: This is a protocol that targets an interesting topic where there is an existing knowledge gap. The design is good and the chosen outcomes relevant. One challenge is that the inclusion strategy, as I understand, will lead to inclusion of those who has not primarily contacted the health care because of a heavy burden of their back pain complaints. This means that the population will include both those with minor and major problems. Patients with different feelings of burden will probably have different chances of improvement of the intervention. This can probably be handled in future analysis taking the disability level into the calculations and elaboration of mediators including disability level.

Response: Thank you for raising these points. The analysis will be adjusted for baseline Oswestry Disability Score. We also plan to carry out a mediation analysis for which a separate protocol is being prepared.

Comment: It will be of great importance to publish the presentation of the treatment in detail as stated in the protocol.

Response: As noted above, in response to the editor's comment, we have only provided a brief description of the intervention in this protocol paper, as we have a written separate paper providing a detailed rationale and description of the intervention in accordance with TIDieR guidelines. This has been submitted to "Physiotherapy" and is currently under peer review. To include a full description of the intervention in this paper would have made it prohibitively long. Some additional details have been added to the text.

VERSION 2 – REVIEW

REVIEWER	Carlo Ammendolia University of Toronto Canada
REVIEW RETURNED	19-Jul-2018

GENERAL COMMENTS	I believe this study outlined in this protocol has already begun recruiting patients so my comments are general in nature and not expected to change the protocol. 1. the authors suggest that a strength of their protocol is that the primary outcome selected is highly applicable to neurogenic claudication. The ODI, the primary outcome unfortunately was found to be inadequately correlated with objective walking ability (see Tomkins-Lane CC, Battie MC, Macedo LG. Longitudinal construct validity and responsiveness of measures of walking capacity in individuals with lumbar spinal stenosis. Spine Journal: 2014;14(9):1936-43) which is the dominant issue in neurogenic claudication. Although the walking section of the ODI is highly correlated to objective walking this measure is a secondary outcome. I do not believe the ODI is not an adequate measure of neurogenic claudication. 2. it is very surprising that 1-2 sessions of physiotherapy for LSS is " best practice" in the UK. LSS is a chronic and often debilitating condition and I assume that is why the main intervention in this
---

	protocol is highly intensive and is likely no match to 1-2 sessions (received by the control). 3.since physiotherapists providing both the control and main intervention can lead to considerable bias....this should be listed as a potential limitation in the protocol (no section on limitations in protocol). this is a very important study. Thank you for allowing me to review the protocol Carlo Ammendolia
--	---

VERSION 2 – AUTHOR RESPONSE

Reviewer comment: I believe this study outlined in this protocol has already begun recruiting patients so my comments are general in nature and not expected to change the protocol.

That is correct. Recruitment is due to be completed by the end of September 2018.

Reviewer comment: The authors suggest that a strength of their protocol is that the primary outcome selected is highly applicable to neurogenic claudication. The ODI, the primary outcome unfortunately was found to be inadequately correlated with objective walking ability (see Tomkins-Lane CC, Battie MC, Macedo LG. Longitudinal construct validity and responsiveness of measures of walking capacity in individuals with lumbar spinal stenosis. Spine Journal: 2014;14(9):1936-43) which is the dominant issue in neurogenic claudication. Although the walking section of the ODI is highly correlated to objective walking this measure is a secondary outcome. I do not believe the ODI is not an adequate measure of neurogenic claudication.

Thank you for your comments. We appreciate your concerns and have revised the “Strengths and Limitations” section.

Reviewer comment: It is very surprising that 1-2 sessions of physiotherapy for LSS is "best practice" in the UK. LSS is a chronic and often debilitating condition and I assume that is why the main intervention in this protocol is highly intensive and is likely no match to 1-2 sessions (received by the control).

The control intervention is 1-3 sessions and is reflective of UK current practice. Unfortunately, the provision of physiotherapy in the UK is variable in this patient group and some patients will receive no or little physiotherapy input. We need to generate high quality evidence regarding both the clinical and cost-effectiveness of different packages of physiotherapy to inform the commissioning and provision of these services.

Reviewer comment: Since physiotherapists providing both the control and main intervention can lead to considerable bias....this should be listed as a potential limitation in the protocol (no section on limitations in protocol).

We have listed this as a potential limitation in the “Strengths and Limitations” section. We have also previously addressed this on page 14 of the text: “At some sites, the same physiotherapist delivers both arms of the trial. Both interventions are delivered according to a manualised protocol to reduce the risk of introducing bias to the study and routine quality assurance checks are conducted.”